# Fabrication of CaCO_3_-Coated Vesicles by Biomineralization and Their Application as Carriers of Drug Delivery Systems

**DOI:** 10.3390/ijms23020789

**Published:** 2022-01-12

**Authors:** Chiho Miyamaru, Mao Koide, Nana Kato, Shogo Matsubara, Masahiro Higuchi

**Affiliations:** Department of Life Science and Applied Chemistry, Graduate School of Engineering, Nagoya Institute of Technology, Gokiso-cho, Show-ku, Nagoya 4668-555, Japan; c.miyamaru.378@nitech.jp (C.M.); m.koide.636@nitech.jp (M.K.); n.kato.575@stn.nitech.ac.jp (N.K.); matsubara.shogo@nitech.ac.jp (S.M.)

**Keywords:** CaCO_3_-coated vesicle, DDS carrier, peptide lipid, mineralization, drug release, under weakly acidic condition

## Abstract

We fabricated CaCO_3_-coated vesicles as drug carriers that release their cargo under a weakly acidic condition. We designed and synthesized a peptide lipid containing the Val-His-Val-Glu-Val-Ser sequence as the hydrophilic part, and with two palmitoyl groups at the *N*-terminal as the anchor groups of the lipid bilayer membrane. Vesicles embedded with the peptide lipids were prepared. The CaCO_3_ coating of the vesicle surface was performed by the mineralization induced by the embedded peptide lipid. The peptide lipid produced the mineral source, CO_3_^2−^, for CaCO_3_ mineralization through the hydrolysis of urea. We investigated the structure of the obtained CaCO_3_-coated vesicles using transmission electron microscopy (TEM). The vesicles retained the spherical shapes, even in vacuo. Furthermore, the vesicles had inner spaces that acted as the drug cargo, as observed by the TEM tomographic analysis. The thickness of the CaCO_3_ shell was estimated as ca. 20 nm. CaCO_3_-coated vesicles containing hydrophobic or hydrophilic drugs were prepared, and the drug release properties were examined under various pH conditions. The mineralized CaCO_3_ shell of the vesicle surface was dissolved under a weakly acidic condition, pH 6.0, such as in the neighborhood of cancer tissues. The degradation of the CaCO_3_ shell induced an effective release of the drugs. Such behavior suggests potential of the CaCO_3_-coated vesicles as carriers for cancer therapies.

## 1. Introduction

Cancer is the leading cause of death for humankind. Therapies for cancer are surgery, drug, radiation, etc. Among these, drug therapies are attracting attention as a method that does not damage the “quality of life” of patients, because it suppresses physical invasion. In recent years, molecular-targeted drugs [1,2] and immune checkpoint inhibitors [3,4] have been developed and are being used in drug therapies. These drugs suppress side effects and achieve high anti-cancer effects; however, these drugs are very expensive. On the other hand, many drugs are abandoned in the development stage due to their significant side effects despite their high efficacy. The drug delivery system (DDS) has been refocused to enable the use of these drugs with prominent side-effects. DDS is a system that delivers the drugs “in a minimum amount”, “at the right time”, and “to the right place”. Polymer micelles [5,6], dendrimers [7,8,9], hydrogels [10,11,12,13], and mesoporous nanoparticles [14,15,16] have been reported as DDS carriers. In addition, vesicles have been used for DDS carriers. Vesicles have the following advantages as DDS carriers. They have a low toxicity and antigenicity because the main component is a lipid. The vesicles can be encapsulated hydrophilic drugs in the internal aqueous phase, and incorporated hydrophobic them in the interior of the bilayer membrane, respectively. It is possible to add a specific recognition ability by embedding proteins and/or peptides [17,18] in the bilayer membrane. That is, vesicles can be obtained with a targeting ability for specific tissues [19,20,21]. Although vesicles have many of the above advantages as DDS carriers, they have the drawback of being unstable in vivo because of metabolically disruption. We have to consider the difference between cancer and normal tissues in the molecular design of the carriers for drug therapies. The new blood vessels around the cancer tissue are incomplete, and carriers that do not erupt from normal blood vessels permeate from the blood vesicle wall and accumulated in the cancer tissue. The effect of this carrier accumulating in the cancer tissue is called the “enhanced permeability and retention (EPR) effect” [22], and is found in carriers with a diameter from 50 to 200 nm [23,24]. In addition, cancer cells grow rapidly and are actively metabolized. Therefore, as lactic acid accumulates, the CO_2_ concentration also increases. Furthermore, because the proton pump operates actively, the pH value around the cancer tissue is relatively lower than that of the normal tissue [25,26,27]. From the above, it is necessary that the DDS carrier in drug therapy for cancer diseases has a size capable of exhibiting the EPR effect and that can release the drug under weakly acidic conditions.

In this paper, we attempted the fabrication of stable CaCO_3_-cated vesicles as DDS carriers for drug therapies in cancer. CaCO_3_ is the main inorganic component of the shells and is not toxic. The important point is that CaCO_3_ is insoluble under physiological pH conditions around normal tissue, but dissolves under weakly acidic conditions in the neighborhood of cancer tissue. We hypothesize that the CaCO_3_-coated vesicles are able to effectively release the drugs, owing to the collapse of the shell as it is dissolved in the vicinity of the cancer tissues. CaCO_3_-coated vesicles were obtained by self-supplied CaCO_3_ mineralization [28,29] using the embedded peptide lipid with the Val-His-Val-Glu-Val-Ser sequence as the catalyst of urea hydrolysis. The vesicles maintained their spherical shape even under a high vacuum; however, the vesicles collapsed easily under the weakly acidic condition owing to the dissolution of the CaCO_3_ shells. Both the encapsulated hydrophobic and hydrophilic drugs were released by dissolving the CaCO_3_ shell under weakly acidic conditions, such as in the neighborhood of cancer tissues.

## 2. Results and Discussion

### 2.1. Structure of CaCO_3_-Coated Vesicles Fabricated by the Peptide Lipids Induced Minerarlzation

#### 2.1.1. CaCO_3_-Coating of the Vesicle Surface

In our previous studies [28,29], we have been reported that the Val-His-Val-Glu-Val-Ser peptide acts as mineral source supplier for CaCO_3_ mineralization through the hydrolysis of urea. We have proposed the mechanism of urea hydrolysis by the Val-His-Val-Glu-Val-Ser peptide as follows. The imidazole group of the His residue activates the hydroxyl group of the Ser residue of the peptides by taking proton [29]. The interaction between His and Ser residues among the Val-His-Val-Glu-Val-Ser peptides is well-known as the “charge relay effect” that is seen in the serine protease [30]. The activated hydroxyl group of the Ser residue hydrolyzes one urea molecule to two ammonium cations, and has one carbonate anion as the mineral source of the CaCO_3_ mineralization. We designed and synthesized the peptide lipid with a hydrophilic Val-His-Val-Glu-Val-Ser sequence as a mineral source supply site, and with two hydrophobic palmitoyl groups as the anchors for embedding in a vesicle membrane. Figure 1 shows the chemical structure of the peptide lipid. In this figure, we show the reaction mechanism of urea hydrolysis by the peptide lipids.

Azolectin vesicles embedded the peptide lipids were prepared. The obtained vesicle with an interior containing 150 mM NaCl, and an exterior containing 100 mM NaCl, 25 mM Urea, and 25 mM Ca(OAc)_2_ in an isotonic condition. Under this condition, we considered that the vesicle remained spherical and the Val-His-Val-Glu-Val-Ser sequence of the peptide lipid presented on the outer surface of the vesicle hydrolyzed urea to form the CaCO_3_ shell. Figure 2 shows the schematic picture of the fabrication of the CaCO_3_-coated vesicle by mineralization under the isotonic condition. In addition, we confirmed that the thermal pyrolysis of urea in an aqueous solution occurred only above 30 °C [29]. Therefore, CaCO_3_ coating on the vesicle outer surface, owing to the mineralization by peptide lipid induced urea hydrolysis, was performed at 20 °C without bulk mineralization.

#### 2.1.2. Structure of CaCO_3_-Coated Vesicles

We observed the morphological changes of CaCO_3_-coated vesicles obtained by the mineralization reaction using transmission electron microscopic (TEM) observations. Figure 3 shows the TEM images of the vesicles after CaCO_3_ mineralization for 8, 11, 18, and 25 days, respectively. For comparison, the TEM image of the vesicle before CaCO_3_ mineralization is shown in Figure 3e. The vesicle before mineralization did not have a clear structure. This implies that the non-coated vesicle was crushed under the high vacuum during TEM observation. From the TEM images after 8 (Figure 3a) and 11 days (Figure 3b), when the mineralization reactions were short, the vesicles were crushed and deformed. Furthermore, many white spots were observed on the vesicle surface after 8 days of mineralization (Figure 3a). This suggests that if the mineralization period is short, uncoated areas with CaCO_3_ occur. On the other hand, when increasing of the mineralization period, the crushed structure disappeared and spherical structures that could maintain those shapes even in vacuo were clearly observed (Figure 3c,d). This suggests that CaCO_3_-coating by mineralization for 18 days fabricates stable vesicles a maintaining spherical structure.

The hydrodynamic diameter of the CaCO_3_-coated vesicles obtained by 18-day mineralization was estimated by dynamic light scattering (DLS) measurements. The diameter of the CaCO_3_-coated vesicle was 193.6 ± 84.4 nm (Figure A1a). This value corresponded to the diameter of the spherical particles observed by TEM, and the size was within the range where the EPR effect could be expected. However, the standard deviation of the radius of the CaCO_3_-coated vesicle was high, and there were coated vesicles of 200 nm or more. The size of the CaCO_3_-coated vesicles could be controlled by adjusting the period of mineralization. In addition, the diameter of the vesicle before mineralization was 122.7 ± 22.7 nm (Figure A1b). This implies that the diameter increased owing to the surface coating by CaCO_3_ mineralization.

For comparison, CaCO_3_ mineralization was performed on the surface of the azolectin vesicle (azolectin/peptide lipid ratio of 200:1) with a lower peptide lipid content. Furthermore, mineralization was also performed on the surface of the dipalmitoylphosphatidylcholine (DPPC) vesicle containing the peptide lipid (DPPC/peptide lipid ratio of 100:1) which has a gel state at 20 °C [31]. In either system, after 18 days of mineralization, the vesicles collapsed under TEM observation, and the spherical structure could not be maintained in vacuo. These results suggest that the Val-His-Val-Glu-Val-Ser sequence in the peptide lipids needs to collide in the vesicle membrane and interact between His and Ser in order to hydrolyze urea. That is, the peptide lipids must exist at a collisional concentration in the fluid vesicle membrane.

The composition distribution of CaCO_3_-coated vesicles obtained after 18-day mineralization was evaluated using energy dispersive X-ray spectrometry (EDX) mapping. Figure 4 shows the elemental mappings of phosphorus (Figure 4a) assigned to azolectin, which is the major component of the vesicle membrane, and calcium (Figure 4b) corresponded to the mineralized CaCO_3_. CaCO_3_ was found only in the vicinity of the vesicle, and formation in the bulk was not observed.

We investigated the crystal phase of the mineralized CaCO_3_ shell on the vesicle using X-ray diffraction (XRD) measurements. CaCO_3_ takes on different crystal phases as most stable calcite phase, semi-stable aragonite phase, and unstable vaterite phase [32]. Figure 5 shows the XRD profile of the CaCO_3_-coated vesicles obtained after the 18-day mineralization. In this figure, the standard profiles of calcite [33], aragonite [34], and vaterite [35] are shown. The XRD profile of the CaCO_3_-coated vesicle was very similar to the standard profile of the calcite phase. We assigned crystal faces of the diffraction peaks shown in Figure 5 from those of 2θ and a relative intensity, along with those of the standard calcite XRD profile (Table A1). The 2θ and relative intensities for the CaCO_3_-coated vesicle were in good agreement with those of calcite. This implies that the mineralized CaCO_3_ on the vesicle surface took the most stable calcite phase. In this system, CaCO_3_ was formed on the vesicle surface through the reaction between Ca^2+^ and CO_3_^2−^. CO_3_^2−^ was supplied by the peptide lipid owing to the urea hydrolysis. On the other hand, Ca^2+^ was captured on the vesicle surface as a result of the electrostatic interaction with the phosphate group of azolectin, and/or the carboxy group of the Glu side chain of the peptide lipid. It is considered that CaCO_3_ took the most stable calcite phase, as there was no clear regularity in the spatial arrangement of phosphate and carboxyl groups on the vesicle surface.

To evaluate the internal structure of the CaCO_3_-coated vesicle obtained after the 18-day mineralization, three-dimensional TEM (3D-TEM) observations and a tomographic analysis were performed. Figure 6 shows the 3D-TEM and tomography image of the CaCO_3_-coated vesicle. We can see that there was a cavity inside the CaCO_3_-coated vesicle. Furthermore, the thickness of the CaCO_3_ shell was ca. 20 nm. These results suggest that the vesicle obtained for the 18-day mineralization had a stable spherical structure whose surface was coated with calcite, while maintaining the intrnal aqueous phase. The CaCO_3_-coated vesicle was useful as the DDS carrier because it had a particle size capable of exhibiting the EPR effect and could encapsulate both hydrophilic and hydrophobic drugs.

### 2.2. Drug Release Properties of CaCO_3_-Coated Vesicles

#### 2.2.1. Dissolution Behaviors of CaCO_3_ Shells

We successfully fabricated the CaCO_3_-coated vesicle as a stable DDS carrier with a particle size capable of exerting an EPR effect. We considered that the CaCO_3_-coated vesicles act as environmentally responsive DDS carriers, with the advantage that the CaCO_3_ shells dissolve under acidic conditions. The strategy for using CaCO_3_-coated vesicles as DDS carriers is follows. First, the CaCO_3_ shell on the vesicle dissolves under the weakly acidic conditions, such as in the neighborhood of cancer tissues. Next, the vesicle lost the CaCO_3_ shell collapses and releases the drugs. We first investigated the dissolubility of CaCO_3_ shell on the vesicle under various pH conditions. The quantification of free Ca^2+^ generated by the dissolution of the CaCO_3_ shell was performed by fluorescence measurements using 8-amino-2-[(2-amino-5-methylphenoxy)methyl]-6-methoxyquinoline-*N,N,N’,N’*-tetraacetic acid, tetrapotassium salt (Quin 2) [36]. We used the CaCO_3_-coated vesicle obtained for the 18-day mineralization. Figure 7 shows the concentration changes in the free Ca^2+^ produced by the dissolution of the CaCO_3_ shell in the solution mimicking the pH environment in the neighborhood of cancer (pH 6.0) and normal (pH 7.4) tissues, respectively. For comparison, the concentration changes in free Ca^2+^ under weakly basic conditions at pH 8.0 are also shown in this figure. The CaCO_3_ shells of the vesicles were gradually dissolved and there was a slight increase in the free Ca^2+^ concentration in the external aqueous phase when the CaCO_3_-coated vesicles were dispersed in the pH 8.0 buffer. Under pH 7.4, mimicking the normal tissue neighborhood, the free Ca^2+^ concentration increased monotonically, although the dissolution rate of the Ca shells was faster than at pH 8.0. On the other hand, the dissolution phenomena of Ca shells in the pH environment at pH 6.0, which mimicked the neighborhood of the cancer tissue, was clearly different from that at pH 8.0 and pH 7.5. Under this condition, the CaCO_3_ shells dissolved rapidly in 2 days, and then the concentration of free Ca^2+^ gradually increased. This result indicates that the shells of CaCO_3_-coated vesicles showed effective dissolubility under the weakly acidic condition as compared with the neutral and weakly basic conditions. That is, the CaCO_3_-coated vesicle had the ability to recognize environmental pH conditions.

#### 2.2.2. Hydrophilic and Hydrophobic Drug Release Properties

Next, we investigated the drug release behavior from the CaCO_3_-coated vesicles under various pH conditions. We used the CaCO_3_-coated vesicles obtained from the 18-day mineralization, which were used for the shell dissolubility measurements. However, the vesicles used in the drug release experiments encapsulated hydrophilic Rhodamine 6G (Rh6G) in the internal aqueous phases or incorporated the hydrophobic Pyrene (Py) in the lipid bilayer membranes. The release of hydrophilic Rh6G was evaluated by permeation from the vesicle interior to the outer aqueous phase as the amount of Rh6G per 1 g of CaCO_3_-coated vesicles (Figure 8a). On the other hand, the release of hydrophobic Py was evaluated by the transfer amount from the bilayer membrane of CaCO_3_-coated vesicles to that of the pure azolectin vesicle as Py amount per 1 g of CaCO_3_-coated vesicles (Figure 8b). The amount of drug release increased with decreasing the pH of the medium in both hydrophilic Rh6G and hydrophobic Py. This propensity was consisted with that of the CaCO_3_ shell dissolubility shown in Figure 7. This means that the drug release from the CaCO_3_-coated vesicle was caused by the dissolution of the shell. On the other hand, the profiles between the transfer of hydrophobic Py and the release of hydrophilic Rh6G were observed to be clearly different under the weakly acidic condition of pH 6.0, mimicking the neighborhood of the cancer tissue. The transfer profile of Py showed the saturated curve (Figure 8b; pH 6.0). However, the release profile of Rh6G at pH 6.0 showed a slower release of up to 3 days. Then, the rapid increase in the amount of Rh6G released was observed, and the release amount reached the equilibrium value after 8 days.

This difference in drug release behavior can be explained as follows. Here, we discuss the initial process of the drug release phenomena as a result of the dissolution behavior of the CaCO_3_ shell. The release of hydrophilic Rh6G from the CaCO_3_-coated vesicle is thought to be caused by the dissolution of the CaCO_3_ shell and the subsequent collapse of the vesicle (Figure 9a). On the other hand, it is thought that the release of hydrophobic Py occurred after contact between the CaCO_3_-coated vesicle and the pure azolectin vesicle (Figure 9b). That is, the rapid transfer of Py was caused in the initial state owing to the immediate contact of the CaCO_3_-coated vesicle to the azolectin vesicle as soon as partial dissolution of the CaCO_3_ shell occurred. This is implied from the rapid rising in the Py transfer profile (Figure 8b; pH 6.0) during the period, when CaCO_3_ shell dissolution occurred rapidly in the first 2 days (Figure 7; pH 6.0). The amount of hydrophilic Rh6G released showed a monotonous increasing tendency (Figure 8a; pH 6.0) during the rapid dissolution of the CaCO_3_ shell, but the release amount increased rapidly after 3 days, during which the concentration of free Ca^2+^ was gradually increased (Figure 7; pH 6.0). The Py transfer occurred through contact of the exposed bilayer membrane of the CaCO_3_-coated vesicle with the azolectin vesicle (Figure 9b). However, the release of hydrophilic Rh6G did not occur until the vesicle disintegrated after the dissolution of the CaCO_3_ shell (Figure 9a). The hydrophilic Rh6G could not permeate the hydrophobic bilayer membrane, so the release of Rh6G required the vesicle disintegration. The hydrophilic drugs were encapsulated in the inner aqueous phase of the vesicle, while the hydrophobic drugs were incorporated in the bilayer membrane. It is thought that drug releases were performed by different mechanisms due to the difference in the location of the drugs.

## 3. Materials and Methods

### 3.1. Materials

#### 3.1.1. Peptide Lipid

The amino acid sequence of the peptide lipid (Figure 1), Val-His-Val-Glu-Val-Ser, was chosen as the hydrolysis site of the urea to produce the CO_3_^2−^, which is the mineral source of CaCO_3_ [28,29]. Lysine was introduced at the *N*-terminal of the peptide. Two palmitoyl groups were attached as the anchor to the vesicle through a condensation reaction between the amino group of *N*-terminal lysine and the carboxyl group of palmitic acid (Nacalai Tesque Inc., Kyoto, Japan). Peptide lipid synthesis was carried out using 9-fluorenylmethoxycarbonyl (Fmoc) chemistry on 4-(2,4-dimethoxyphenyl-Fmoc-aminomethoxyl)phenoxyacetyl-norleucine loaded cross-linked ethoxylate acrylate (CLEAR-amide) resin [37]. Fmoc-amino acids (Fmoc-Val, Fmoc-His(Trt), Fmoc-Glu(OBut)-H_2_O, and Fmoc-Ser(But)) and clear-amide resin were purchased from Peptide Institute, Inc. (Osaka, Japan). The Fmoc-amino acid for *N*-terminal lysine residue was Fmoc-Lys(Fmoc), which was purchased from Watanabe Chemical Industrials, Ltd. (Hiroshima, Japan). Two palmitoyl groups were attached at the *N*-terminal amino group and side chain of the lysine residue by the same chemistry. Peptide lipid cleavage and deprotection of the side chain protecting groups were performed at the same time to obtain the peptide lipid. The obtained peptide lipid was identified by matrix-assisted laser desorption/ionization time-of-flight mass spectroscopy (MALDI-TOF-MS) on a JEM-S3000 system (JEOL Ltd., Tokyo, Japan). Molecular weights of the synthesized peptide lipid were obtained to be 1272.9 and 1294.9 from the MALDI-TOF-MS measurement (Figure A2). The calculated molecular weight of the peptide lipid was 1272.7. The molecular weights obtained by the MALDI-TOF-MS measurement were in fair agreement with the calculated values as [M+H]^+^ (m/z = 1273.7) and [M+Na]^+^ (m/z = 1295.7), respectively. From the MALDI-TOF-MS study, we obtained evidence indicating the successful synthesis of the designed peptide lipid.

#### 3.1.2. Vesicles

The peptide lipid was dissolved in 2,2,2-trifluoroethanol (TFE; Nacalai Tesque Inc., Kyoto, Japan). Azolectin (Nacalai Tesque Inc., Kyoto, Japan) was dissolved in chloroform (Nacalai Tesque Inc., Kyoto, Japan). These solutions were mixed and poured into a glass flask and then a thin film was formed on the interior surface of the flask from the evaporation of the solvents. The molar ration of the peptide lipid to azolectin was fixed at 0.01. The molecular weight of azolectin used was that of dioleoylphosphatidylcholine, 786. An aqueous solution containing 150 mM NaCl (Nacalai Tesque Inc., Kyoto, Japan) was added to this flask, and was sonicated by Branson Sonifier 250 (Danbury, CT, USA) for 10 min, under a nitrogen atmosphere at 0 °C. The pH of the vesicle dispersion was adjusted to pH 7.4. For comparison, a dipalmitoylphosphatidylcholine (DPPC; FUJIFILM Wako Pure Chemical Co., Osaka, Japan) vesicle, which is in a gel state at room temperature, was also prepared in the same manner.

We used model drugs rhodamine 6G (Rh6G; Nacalai Tesque Inc., Kyoto, Japan), as a hydrophilic drug, and pyrene (Py; FUJIFILM Wako Pure Chemical Co., Osaka, Japan), as a hydrophobic drug. Encapsulation of these drug models in the vesicles was performed as follows: For the encapsulation of the hydrophilic drug model, Rh6G was added to the aqueous solution containing 150 mM NaCl. The Rh6G encapsulated vesicle was obtained by sonication in the aqueous solution containing Rh6G. The concentration of Rh6G in the 150 mM NaCl aqueous solution was 1 mM. On the other hand, to introduce the hydrophobic model drug, Py, into the bilayer membrane of the vesicle, Py was mixed into the thin film consisting of the peptide lipid and azolectin. Py and azolectin were dissolved in chloroform. The molar ration of the Py to azolectin was 0.12. The TFE solution containing the peptide lipid was added to the chloroform solution in the glass flask and then the thin film containing Py was formed through the evaporation of the solvents. An aqueous solution containing 150 mM NaCl was added to the flask and was sonicated to prepare the Py incorporated vesicle.

#### 3.1.3. CaCO_3_ Coating on the Vesicle Surface by Mineralization

CaCO_3_ coating on the vesicle surface was performed by mineralization using CO_3_^2−^ generated by urea hydrolysis induced by the Val-His-Val-Glu-Val-Ser sequence of the peptide lipid [28]. Aqueous solutions containing 150 mM urea (Nacalai Tesque Inc., Kyoto, Japan) and 150 mM calcium acetate (Nacalai Tesque Inc., Kyoto, Japan) were added to the vesicle dispersions either with or without the model drugs obtained above. The volume ration of the vesicle dispersion, aqueous solutions of urea, and calcium acetate was 4:1:1. Under this condition, the inner aqueous phase of the vesicle contained 150 mM NaCl, and the outer consisted of 100 mM NaCl, 25 mM urea, and 25 mM calcium acetate. The obtained vesicle dispersion was an isotonic system in which no osmotic pressure was generated between the inside and outside of the bilayer membrane. The vesicle dispersions were gently shaken at 20 °C, after which the pyrolysis of urea did not occur [29]. After 18 days of mineralization, the vesicle dispersions were centrifuged at 10,000 rpm for 20 min. The precipitates were re-dispersed in water adjusted to pH 8.0, and were centrifuged to obtain the precipitates again. This centrifugation and redispersion cycle were performed twice more to remove the unreacted calcium acetate, urea, and the unincluded model drugs from the vesicle outer aqueous phase. The vesicle dispersions were lyophilized, and the vesicles with or without model drugs were stored frozen until they were used in the experiments.

### 3.2. Methods

#### 3.2.1. Transmission Electron Microscopic Observations

The morphologies of the CaCO_3_-coated vesicles obtained by mineralization were determined using a scanning transmission electron microscope (STEM; JEM-z2500, JEOL Ltd., Tokyo, Japan) equipped with an Ultra Scan CCD camera (US1000; Gatan Inc., Pleasanton, CA, USA) in TEM mode. Elemental analysis and mappings of the CaCO_3_-coated vesicles were performed in STEM mode equipped with an energy dispersive X-ray spectrometer (EDX; EX-37001, JEOL Ltd., Tokyo, Japan). An aliquot of the vesicle dispersion obtained after various mineralization periods was placed on an elastic carbon-coated STEM grid, and the CaCO_3_-coated vesicles were allowed time to adsorb onto its surface. After the adsorption, the excess solution was removed by absorption onto filter paper, and the grid was rinsed with water to remove the unreacted urea and calcium acetate.

Three-dimensional TEM tomography was performed to analyze the internal structure of the CaCO_3_-coated vesicle. Projection images under sample rotation angles from −60° to +60° (at 1° increments) were automatically acquired. The projection images were reconstructed into 3D images using a TEM tomography system (TEMography, SYSTEM IN FRONTIER Inc., Tokyo, Japan). TEM and STEM observations were performed using unstained samples at an acceleration voltage of 200 kV.

#### 3.2.2. Dynamic Light Scattering Measurements

The hydrodynamic diameters of the CaCO_3_-caoted vesicles were evaluated by dynamic light scattering (DLS) measurement. The lyophilized CaCO_3_-coated vesicle obtained after 18 days of mineralization was re-dispersed in water adjusted to pH 7.4 to prepare a measurement sample. The concentration of the CaCO_3_-coated vesicle was 0.1 mg/mL. The DLS measurements were performed at 25 °C using a Zetersizer ZS (Malvern Panalytical Ltd., Cambridge, UK). For comparison, DLS measurements were also performed on CaCO_3_-uncoated vesicles before mineralization.

#### 3.2.3. X-ray Diffraction Measurements

The crystal structure of the CaCO_3_ shell formed on the vesicle surface was investigated by X-ray diffraction (XRD) measurement. The lyophilized CaCO_3_-coated vesicle obtained after 18 days of mineralization was used. XRD measurements were carried out using a SmartLabSE (Shimazu Co., Kyoto, Japan) equipped with a 1.8 kW CuKα ceramic X-ray tube operating at 40 kV and 30 mA.

#### 3.2.4. Dissolution Behaviors of Shells on the CaCO_3_-Coated Vesicles

We investigated the dissolubility of the CaCO_3_ shell on the vesicle surface under various pH conditions. The lyophilized CaCO_3_-coated vesicle obtained after 18 days of mineralization was re-dispersed in 100 mM 2-[4-2-(hydroxyethyl)-1-piperazinyl]ethanesulfonic acid (HEPES, Nacalai Tesque Inc., Kyoto, Japan)–Tris(hydroxylmethyl)aminomethane (Tris, Nacalai Tesque Inc., Kyoto, Japan) buffers adjusted to pH 6.0, 7.4, and 8.0, respectively. The concentration of the CaCO_3_-coated vesicle was 0.01 g/L. The CaCO_3_-coated vesicle dispersion at each pH condition was gently shaken at 36 °C. We measured the concentration changes of the Ca^2+^ generated by the dissolution of the CaCO_3_ shell. The quantification of free Ca^2+^ was performed by fluorescence measurements using 8-amino-2-[(2-amino-5-methylphenoxy)methyl]-6-methoxyquinoline-*N,N,N’,N’*-tetraacetic acid, tetrapotassium salt (Quin 2, Dojindo Co. Ltd., Kumamoto, Japan) as a quantification reagent for Ca^2+^. At regular intervals, 1 mL of the vesicle dispersion was collected from the dispersion of each pH condition, and 20 μL of 1.44 mM Quin 2 aqueous solution was added, and then fluorescence measurements were carried out using a spectrofluorophotometer (RF-5300, Shimadzu Co., Kyoto, Japan). The excitation wavelength of Quin 2 was 339 nm, and the fluorescence was observed at 492 nm. We created the calibration curve of the relationship between the Ca^2+^ concentration and Quin 2 fluorescence intensity in each buffer solution. The Ca^2+^ concentrations were calculated from the obtained fluorescence intensities, based on the calibration curves.

#### 3.2.5. Drug Release Experiments

We performed drug release measurements of the hydrophilic model drug, Rh6G, from the CaCO_3_-coated vesicle. The encapsulated Rh6G in the vesicle interior was quenched, and only the Rh6G released from the vesicle emitted fluorescence. The amounts of drug released were determined from the fluorescence intensity of Rh6G in the external aqueous phase. The CaCO_3_-coated vesicle encapsulated Rh6G obtained after 18 days of mineralization was re-dispersed in 100 mM HEPES–Tris buffers adjusted to pH 6.0, 7.4, and 8.0, respectively. The concentration of the CaCO_3_-coated vesicle was 0.01 g/L. The CaCO_3_-coated vesicle dispersion at each pH condition was gently shaken at 36 °C, and the intensity changes of Rh6G fluorescence were monitored on a spectrofluorophotometer. The excitation and emission wavelength of Rh6G were 527 and 551 nm, respectively. The amounts of drug release under each pH condition were calculated based on the calibration curve obtained in each buffer solution.

Furthermore, we investigated the release of the hydrophobic model drug, Py, from the CaCO_3_-coated vesicle. We evaluated the transfer amounts of Py from the CaCO_3_-coated vesicle to the pure azolectin vesicle by the fluorescence measurements. The pure azolectin vesicles, which were reservoirs of released Py from the CaCO_3_-coated vesicles, were prepared by sonication in buffer solutions adjusted to pH 6.0, 7.4, and 8.0, respectively. The concentration of pure azolectin vesicles was 0.1 wt%. Then, 1 mg of the CaCO_3_-coated vesicle containing Py obtained after 18 days of mineralization was weighted into a microtube. The pure azolectin vesicle dispersions adjusted at pH 6.0, 7.4, and 8.0 were added to the microtubes, and then the microtubes were each shaken at 36 °C. At regular intervals, the dispersions were centrifuged at 10,000 rpm for 10 min, and the azolectin vesicles with transferred Py were collected as a supernatant. The amounts of transferred Py under each pH condition were determined by the fluorescence measurement of the collected supernatant, based on the calibration curve of the vesicle containing 1 wt% Py obtained in each buffer solution. The excitation and emission wavelength of Py were 342 and 375 nm, respectively.

## 4. Conclusions

In this study, we attempted to fabricate a new DDS carrier especially useful for cancer therapy. The vesicle surface was coated with CaCO_3_ owing to the self-supplied mineralization induced by the embedded peptide lipid. The CaCO_3_-coated vesicle maintained its spherical shape even under a high vacuum; however, the vesicle collapsed easily under the weakly acidic condition, mimicking the neighborhood of the cancer tissue, owing to the dissolution of the CaCO_3_ shell. It was also confirmed that the CaCO_3_-coated vesicle could encapsulate both the hydrophilic and hydrophobic drugs as the DDS carrier. The encapsulated drugs, hydrophilic Rh6G and hydrophobic Py, were released effectively through the dissolution of the CaCO_3_ shell under weakly acidic conditions such as the neighborhood of cancer tissues. The release of these drugs behaved differently depending on the location of the drug in the vesicle. The obtained CaCO_3_-coated vesicle is expected to be an effective DDS carrier in cancer drug therapy.

## Figures and Tables

**Figure 1 ijms-23-00789-f001:**
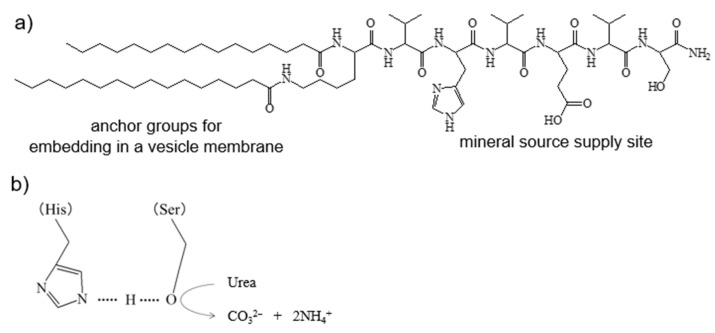
(**a**) Chemical structure of the peptide lipid. (**b**) Hydrolysis mechanism of urea by charge relay between His and Ser.

**Figure 2 ijms-23-00789-f002:**
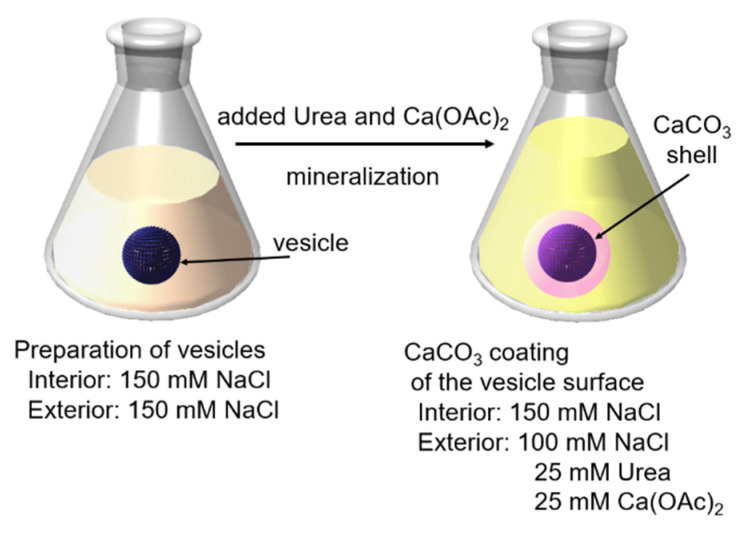
Schematic picture of the fabrication of the CaCO_3_-coated vesicle by the self-supplied mineralization under the isotonic condition.

**Figure 3 ijms-23-00789-f003:**
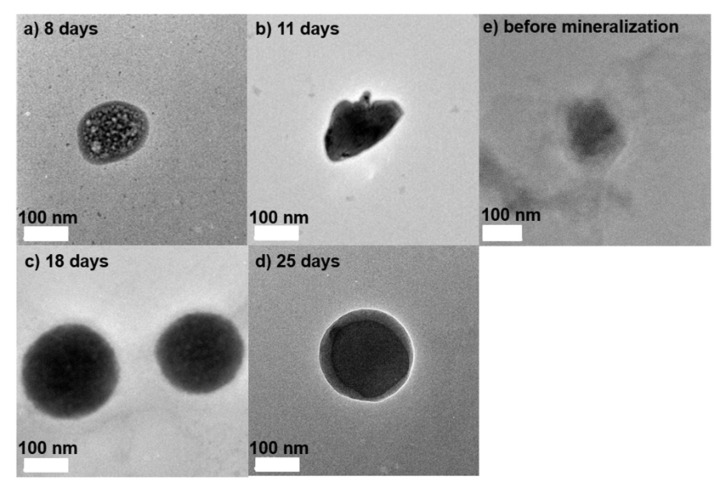
TEM images of vesicles after CaCO_3_ mineralization for (**a**) 8 days, (**b**) 11 days, (**c**) 18 days, and (**d**) 25 days. (**e**) TEM image of the vesicle before mineralization.

**Figure 4 ijms-23-00789-f004:**
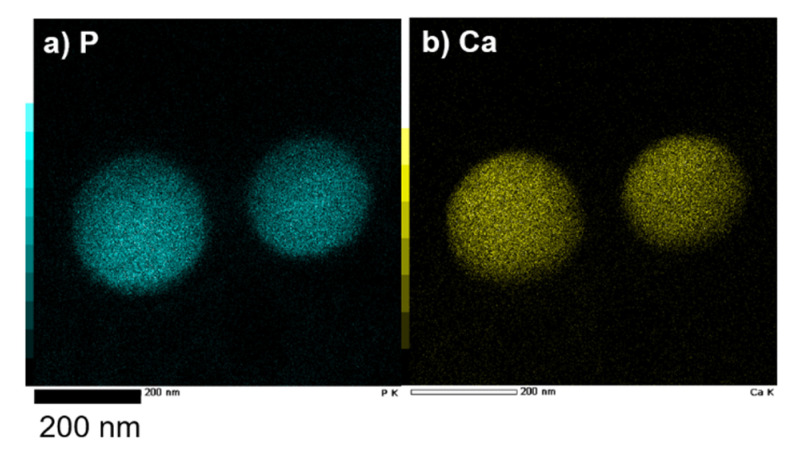
EDX mapping images of (**a**) phosphorous and (**b**) calcium for CaCO_3_-coated vesicles after 18-day mineralization. “P” and “Ca” indicate phosphorus of azolectin and calcium of CaCO_3_, respectively.

**Figure 5 ijms-23-00789-f005:**
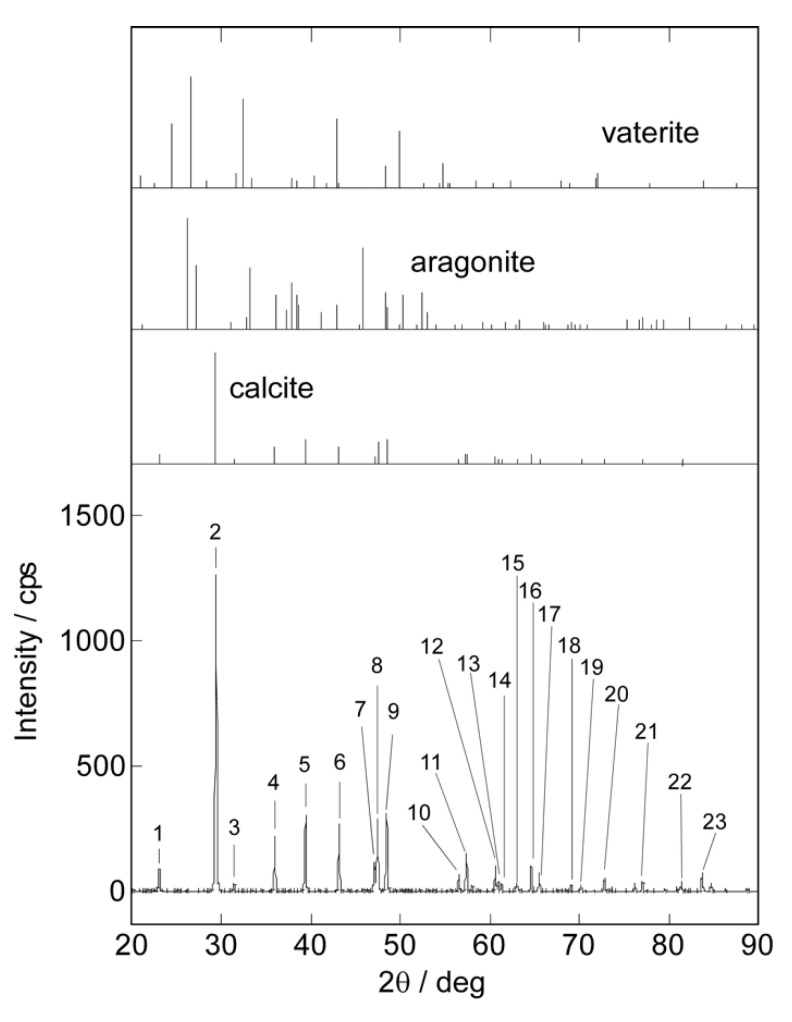
XRD profile of CaCO_3_-coated vesicles after mineralization for 18 days. Standard profiles of calcite, aragonite, and vaterite are shown in this figure, respectively. The assignment of each diffraction peak is shown in Table A1.

**Figure 6 ijms-23-00789-f006:**
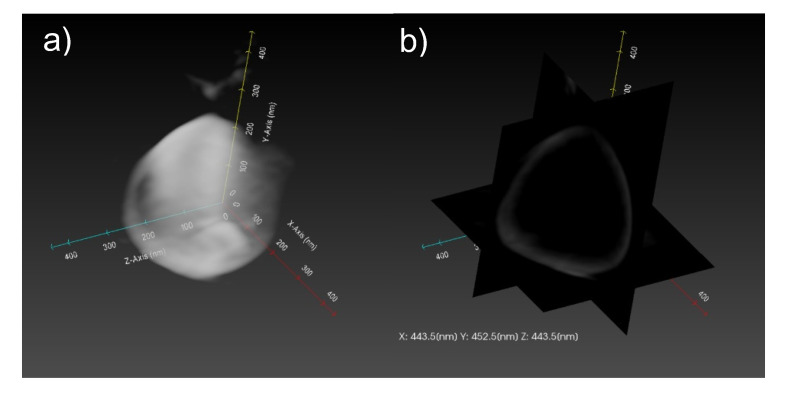
(**a**) 3D-TEM and (**b**) tomography images of CaCO_3_-coated vesicle for 18-day mineralization. Axis length information; *X* axis: 443.5 nm; *Y* axis: 452.5 nm; *Z* axis: 443.5 nm.

**Figure 7 ijms-23-00789-f007:**
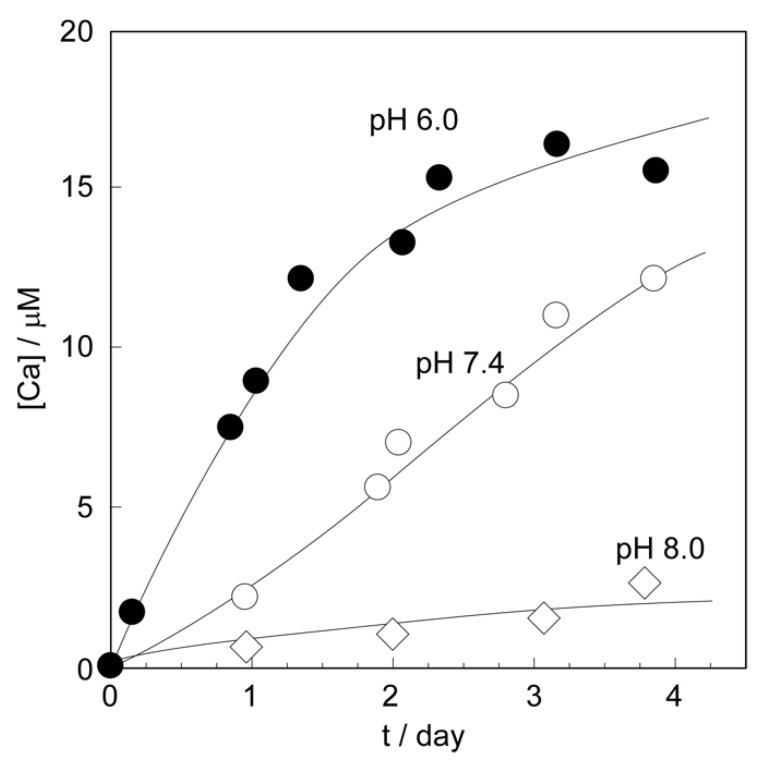
The concentration changes in free Ca^2+^ produced by the dissolution of the CaCO_3_ shell under various pH conditions of pH 8.0, pH 7.4, and pH 6.0, respectively.

**Figure 8 ijms-23-00789-f008:**
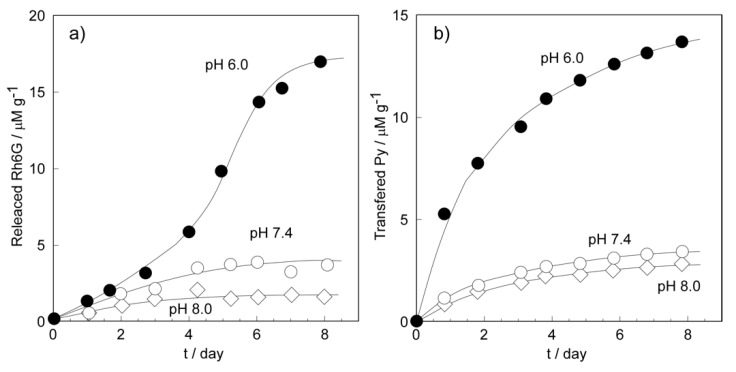
(**a**) The release profile of the hydrophilic Rh6G from the vesicle interior to the outer aqueous phase, and (**b**) the transfer of hydrophobic Py from the bilayer membrane of CaCO_3_-coated vesicles to that of the pure azolectin vesicle under the various pH conditions.

**Figure 9 ijms-23-00789-f009:**
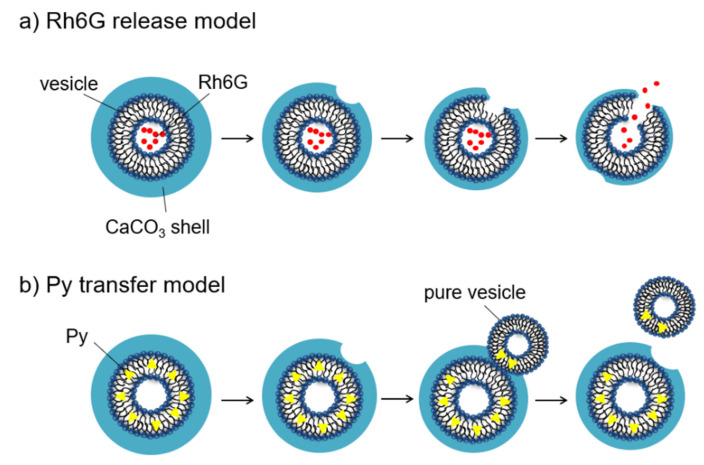
Schematic pictures of the drug release mechanism. (**a**) Release of hydrophilic Ph6G as a result of the collapse of the vesicle. (**b**) Transfer of hydrophobic Py as a result of contact with the vesicle.

## Data Availability

Not applicable.

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
