# Peer review of "Fabrication of CaCO3-Coated Vesicles by Biomineralization and Their Application as Carriers of Drug Delivery Systems"

_ijms, 2022, doi:10.3390/ijms23020789_

Round 1
Reviewer 1 Report
The authors described e method how to fabricate CaCO3-coated vesicles as drug delivery system carriers for the drug therapies in cancer diseases. The vesicles has been morphologically characterized by TEM and 3D TEM while the CaCo3 mineralization had been evaluated by the X-ray diffraction. The author also studied the incapsulation and the release of hydrophobic and hydrophilic drugs
The paper is well written and could be of interest for the reader of IJMS but needs some minor revisions
1) I suggest to revised the title making explicit the acronym SSD. The same in the first line of the abstract
2) In the paragraph 2.1 has been explain how to fabricate the CaCO3 coated vesicles. I suggest to add a scheme describing the procedure and the reaction involved in the preparation. This should help the reader to understand all the pipeline of the fabrication. In addition, it is not clear why the hydrolysis of the urea permit the formation of the CaCO3
3) To confirm all the hypotheses argued by the authors and to increase the quality of the manuscript I suggest to add a panel in figure 2 with a TEM image of an uncoated vesicle
4) After the coating the dimensions of the vesicles have a big increase of variability the author should discuss this point. The size of the coated particles is near the high limit of the EPR (50 to 200 nm) and with such high standard deviation most of the vesicles will be higher than 200 nm
5) In figure 3 I suggest to show also a color code map with the overlapping of Ca and P. If the calcium only coat the vesicles you should see a circle (of about 20 nm) where Ca and P are not overlapped. Please remove the semicolons after the letter in the picture.
6) I suggest to increase the dimension of the figure 4 because the dimension of the reference are too small. In addition please put the calcite spectra in place of vaterite and vice versa.
7) Figure 5 need to be improved. Panel b is too dark please set in a different way the contrast. In addition it is not so clear the presence of a cavity as suggested by the author. I suggest to reconstruct the images and to show also half a vesicle to clarify the presence of the cavity.
Author Response
Response for Reviewer 1
Thank you very much for the reviewer comments. We revised the manuscript according to the comments of the reviewers.
Our alterations as a result of the comments of the reviewers are:
1) According to your comment, we revised the title and first line of abstract to define the DDS. And we wrote it in full form.
2) The charge relay effect between the His and Ser residue of the peptide lipids activated the hydroxy group of Ser and urea was hydrolyzed to produce CO32-. The Ca2+ captured on the vesicle surface by the electrostatic interaction with the phosphate group of azolectin, and/or carboxy group of Glu side chain of the peptide lipid. The produced CO32- reacted with captured Ca2+ to mineralize CaCO3. According to your comment, we added reaction scheme of urea hydrolysis in Figure 1. Furthermore, the schematic picture for fabrication of the CaCO3-coated vesicle was added as Figure 2, according to your comment. According to this the numbers in the following figures were shifted.
3) The TEM image of vesicle before CaCO3 mineralization was not described because it was crushed under the high vacuum of TEM observation and the contrast was poor. However, as you pointed out, we describe it for comparison in Figure 3e. In addition, we added a description concerning this.
4) As you pointed out, the average hydrodynamic radius of coated vesicles is at the upper limit of the EPR effect size. In addition, the standard deviation of the radius is high, and there are particles of 200 nm or more. We believe that the size can be controlled by adjusting the period of mineralization. We added a description concerning this.
5) The spatial resolution of EDX mapping is not high. We tried overlapping the Ca and P, but we could not clearly see the non-overlapping part. According to your comment, we removed the semicolons after the letter in the picture.
6) As you pointed out, we increased the size of the standard profiles. In addition, we placed the calcite standard profile near the measured profile.
7) According to your comment, we reconstructed the Figure 9. A cross section of the vesicle was shown to clearly show the bilayer membrane and the internal aqueous phase. In addition, we clarified the existence of the cavity we are suggesting.
I would very much appreciate the valuable suggestions of you again.
Reviewer 2 Report
In this work, Miyamaru et al. have presented an interesting approach to fabricate the CaCO3-coated vesicles for drug delivery applications. Generally, the paper is well written and the data are clearly presented. However, the authors should address the following major concerns before publication:
1) In figure 2, The author presented the TEM analysis of CaCO3-coated vesicles with different mineralization times (8, 11, 18, and 25 days). However, the structure of the uncoated vesicle has not been demonstrated. The TEM analysis of the non-coated vesicles should be also presented to validate the impact of the mineralization time.
2) In section 2.1.2, the author explained that at the short mineralization of 8 or 11 days, the vesicles were crushed and deformed, while with 18 days or more, the shape of the vesicle remained spherical. In my understanding, if the vesicles were deformed on day 11 of the mineralization, how could they remain spherical on day 18? Please clarify this point.
3) In section 2.1.1 the author mentioned about “Azolecithin vesicles”. What is the difference between “azolecithin vesicles” and “azolectin vesicles”?
4) The main purpose of CaCO3-coated vesicles is to be used as DDS carriers for cancer treatment. However, in this work, no biological studies related to the interaction of the CaCO3-coated vesicles are presented. In vitro study on the cytotoxicity and cellular uptake of CaCO3-coated vesicles treated with cancer cells are necessary to observe the interaction of the vesciles with the cancer cells.
Other minor comments:
- In the abstract, please define what is “DDS”. Also, the term "DDS" in the title should be written in full form.
- Please change the figure numbers in section 2.2.2. They should be figure 7 instead of 6.
- In figure 3 caption, “P” and “Ca” should be assigned for phosphorus and calcium, as EDX analysis identifies the elemental composition of materials.
- Please use only “Ca2+” or only “Ca ion” for better consistency.
Author Response
Response for Reviewer 2
Thank you very much for the reviewer comments. We revised the manuscript according to the comments of the reviewers. We would like the language editing by MDPI.
Our alterations as a result of the comments of the reviewers are:
1) The TEM image of vesicle before CaCO3 mineralization was not described because it was crushed under the high vacuum of TEM observation and the contrast was poor. However, as you pointed out, we describe it for comparison in Figure 3e. In addition, we added a description concerning this.
2) In an aqueous solution, the vesicle retains a spherical shape because it is under the isotonic condition during any mineralization period. The vesicle collapse occurred under high vacuum during TEM observation. As shown in experimental section, 3.2.1, TEM observations were performed by collecting samples at regular intervals from the reaction mixture during the mineralization. It is not that the deformed vesicle has regained its spherical shape.
3) The “azolecithin vesicles” is an error of “azolectin vesicles”. We corrected it to “azolectin vesicles”.
4) As you pointed out we think that in vitro studies on cytotoxicity and cellular uptake of CaCO3-coated vesicles are important. In this paper, we discussed the fabrication of the CaCO3-coated vesicle, the dissolution of the shell, and accompanying drug release properties under the weakly acidic condition. In the next study, we are planning the in vitro research you pointed out.
Other minor comments:
・According to your comment, we revised the title and first line of abstract to define the DDS. And we wrote it in full form.
・According to your comment, we revised the caption of Figure 4. “P” and “Ca” was assigned for phosphorus of azolectin and calcium of CaCO3, respectively.
・The schematic picture for fabrication of the CaCO3-coated vesicle was added as Figure 2. According to this the numbers in the following figures were shifted. According to your comment, Figure 6 in section 2.2.2. was changed as Figure 8.
・According to your comment, we unified “Ca ion” to “Ca2+”.
I would very much appreciate the valuable suggestions of you again.
Round 2
Reviewer 2 Report
In the revised manuscript, the authors have addressed all my questions and concerns. Thus, I recommend it for publication in IJMS.